# Teaching Language Models to Hallucinate Less with Synthetic Tasks

**Erik Jones,**[1][‡] **Hamid Palangi,**[2] **Clarisse Simões,**[2] **Varun Chandrasekaran,**[3][‡]
**Subhabrata Mukherjee,**[4][‡] **Arindam Mitra,**[2] **Ahmed Awadallah,**[2] **Ece Kamar**[2]

[1] UC Berkeley  [2] Microsoft Research  [3] UIUC  [4] Hippocratic AI

## Abstract

Large language models (LLMs) frequently hallucinate on abstractive summarization tasks such as document-based question-answering, meeting summarization, and clinical report generation, even though all necessary information is included in context. However, optimizing LLMs to hallucinate less on these tasks is challenging, as hallucination is hard to efficiently evaluate at each optimization step. In this work, we show that reducing hallucination on a *synthetic task* can also reduce hallucination on real-world downstream tasks. Our method, SYNTRA, first designs a synthetic task where hallucinations are easy to elicit and measure. It next optimizes the LLM's system message via prefix-tuning on the synthetic task, and finally transfers the system message to realistic, hard-to-optimize tasks. Across three realistic abstractive summarization tasks, SYNTRA reduces hallucination for two 13B-parameter LLMs using only a synthetic retrieval task for supervision. We also find that optimizing the system message rather than the model weights can be critical; fine-tuning the entire model on the synthetic task can counterintuitively *increase* hallucination. Overall, SYNTRA demonstrates that the extra flexibility of working with synthetic data can help mitigate undesired behaviors in practice.

## 1 Introduction

Large language models (LLMs) are prone to hallucinate—i.e., fabricate entities, details, or other content—when generating responses to queries, even when all salient information is included in context. For example, LLMs can make up citations (Liu et al., 2023), add titles when generating biographies (Min et al., 2023), or invent new product attributes when advertising (Koto et al., 2022). To confidently deploy LLMs, we need methods to monitor and reduce these hallucinations.

Unfortunately, directly reducing LLM hallucinations on real-world tasks is challenging, in part because we cannot scalably evaluate hallucination during optimization. To exhibit this evaluation challenge, suppose the LLM generates the fictional noun "fixed liability response" when summarizing a meeting. Cheap-to-run rule-based manual verifiers would struggle to (i) decide that this term is worth checking, and (ii) identify whether the term is real or fabricated. On the other hand, using humans or LLMs to detect hallucination in long-form outputs is slow, expensive, and error-prone (Guerreiro et al., 2023). It is thus difficult to directly optimize against LLM hallucinations with gradient descent or reinforcement learning, as we cannot efficiently evaluate the loss or reward.

In response, we introduce SYNTRA, a method that uses synthetic data to reduce LLM hallucinations (Figure 1). SYNTRA first designs a *synthetic task* where hallucination can be efficiently and tractably evaluated. It then exploits this tractability by optimizing the LLM system message on the synthetic task via prefix-tuning (Li & Liang, 2021), and finally transfers the system message to realistic tasks.

The core component of SYNTRA is the design of the synthetic task, which provides the only direct supervision signal that captures hallucination. For this supervision signal to be sufficient, we argue that the synthetic task must at least satisfy two properties: (i) LLMs should hallucinate frequently on the task, and (ii) hallucination can be cheaply and automatically evaluated on the task. The former ensures that optimizing on the synthetic task teaches the model to hallucinate less, while the latter

---

[†]Correspondence to `erjones@berkeley.edu` and `hpalangi@microsoft.com`
[‡]Work done at Microsoft Research.

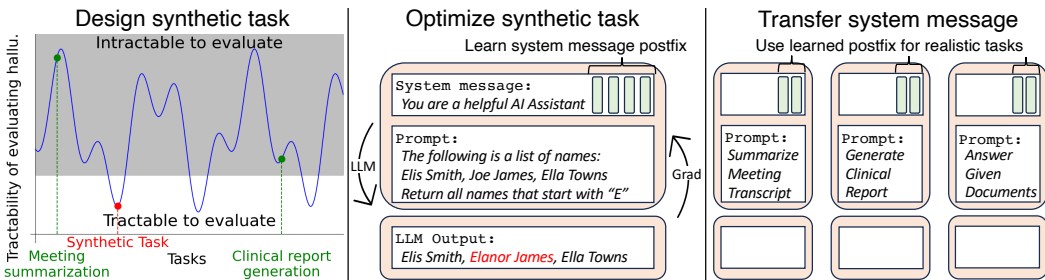

Figure 1: Overview of the SYNTRA framework. We first define a synthetic task where hallucination is easy to tractably evaluate. Next, we optimize the LLM system message on this task by learning a continuous postfix via prefix-tuning. We then transfer the learned system message across real tasks.

makes optimization tractable. Throughout this work, we use the *names retrieval task* as the synthetic task: given a list of random names, we prompt the LLM to retrieve the first $n$ names on the list that start with some letter. We say the model hallucinates if it generates a name that is not on the list.

SYNTRA then optimizes the LLM to reduce hallucination on the synthetic task. To do so, we optimize the LLM system message (e.g.,"*You are a helpful AI assistant*") by appending a continuous postfix to it, then optimizing the postfix. We optimize the system message rather than the whole model to learn high-level instructions for how to hallucinate less, which we expect to transfer well. We additionally optimize the LLM to keep its output constant on a set of reference prompts, so the LLM does not latch onto spurious attributes of the synthetic task.

We find that SYNTRA consistently reduces hallucination across models, tasks, and metrics. We evaluate Vicuna v1.1 (Chiang et al., 2023) and Orca (Mukherjee et al., 2023) on three realistic tasks: search-and-retrieve, meeting summarization, and clinical report generation. Following, Yue et al. (2023), we measure hallucination with GPT-4, and find that optimizing the system message consistently reduces hallucination: on Orca, SYNTRA reduces the hallucination rate by over 7 points on average and 16 points on specific tasks. In contrast, fine-tuning the whole model can *increase* the hallucination rate. We also test whether SYNTRA reduces hallucination according to other metrics, and find its outputs (i) overlap more with references and (ii) contain fewer ungrounded entities.

SYNTRA is not without limitations; it requires designing a synthetic task, and reduces hallucination on some models more than others. Nevertheless our work provides encouraging evidence that we can use synthetic data to isolate and optimize against undesired LLM behaviors.

## 2 THE SYNTRA PIPELINE

We describe SYNTRA (Synthetic Transfer), our method to reduce hallucination on abstractive summarization tasks using synthetic data. We outline the setup (Section 2.1), introduce synthetic tasks (Section 2.2), then describe how SYNTRA reduces hallucination on realistic tasks (Section 2.3).

### 2.1 SETUP

We study LLMs that take prompts (e.g., "*Write a clinical report about the following dialog...*") and produce long-form outputs (e.g., "*Overview: Patient has...*"). The behavior of the LLM is modulated by two features: a system message that provides high-level instructions (e.g., "*You are a helpful and honest AI assistant*)", and the raw model weights. We thus represent the LLM as $LLM_{\phi,\theta}$, a composition of the *system function $s_\phi$* that appends the prompt to the system message $\phi$,[1] and the *raw LLM $f_\theta$* that generates text based on weights $\theta$. Formally, $LLM_{\phi,\theta} = f_\theta \circ s_\phi$. This decomposition into system messages and prompts is standard for many instruction-tuned LLMs (OpenAI, 2023).

We measure hallucination on *abstractive summarization tasks*, where models respond to queries given some context. We model an abstractive summarization task $\tau$ (e.g., generate clinical reports

---

[1]While the system prompt $\phi$ is usually text, this is challenging to optimize. Following Li & Liang (2021) we append continuous embeddings to the text, which can be optimized via gradient descent.

from dialogues) as a distribution over prompts $\mathcal{D}_\tau$, where each prompt $p$ can be decomposed into a query $q$ that is appended to context $c$, i.e., $p = c \parallel q$. For example, the context $c$ might be a dialog between a patient and a doctor, and the query might be "*Write a clinical report containing ...*". We assume that the context $c$ always has all required information to correctly respond to query $q$.

Hallucinations in the abstractive summarization setting are *grounding errors*: errors where the output contains some information that is not supported by the context. We define the hallucination function $h$, which takes in a prompt and output and returns 1 if the output is supported by the prompt, and 0 otherwise. The hallucination rate $\ell_{\text{hal}}$ on task $\tau$ for model parameters $\theta$ and system prompt $\phi$ is

$$\ell_{\text{hal}}(\phi, \theta; \tau) := \mathbb{E}_{p \sim \mathcal{D}_\tau} \; h(p, LLM_{\phi,\theta}(p)), \tag{1}$$

where the expectation is taken over the distribution $\mathcal{D}_\tau$ and sampling during LLM decoding.

The hallucination function $h$ is at best expensive to evaluate for many different user prompts and outputs, and at worse intractable. This is especially true for abstractive summarization, as even humans can struggle to read and remember long contexts while evaluating generated summaries (Guerreiro et al., 2023). To measure hallucination in general, we rely on proxy metrics $h_{\text{prox}}$ that can use reference outputs or external databases to approximate $h$. Existing work has considered many possible proxy metrics including comparing to ground truth outputs (Nan et al., 2021), decomposing outputs into atomic facts and evaluating them separately (Min et al., 2023), using entailment models (Roit et al., 2023), or using LLM judges (Yue et al., 2023; Gao et al., 2023a).

## 2.2 SYNTHETIC TASKS

A natural approach to reduce hallucination is to optimize against the hallucination rate $\ell_{\text{hal}}$ over all tasks simultaneously, but this is intractable; the hallucination function $h$ is expensive to evaluate at scale, which rules out direct optimization. Optimizing against proxy metrics $h_{\text{prox}}$ is also problematic, as they may (i) unfaithfully capture hallucination and (ii) still be expensive to optimize.

Rather than trying to optimize a proxy in general, we instead optimize the true hallucination rate exactly on a carefully constructed *synthetic task*. Concretely, we construct a synthetic task $\tau_{\text{syn}}$ such that the hallucination function restricted to this task, $h\big|_{\mathcal{D}_{\tau_{\text{syn}}}}$, is easy to automatically evaluate and optimize. For example, suppose the prompts contain random first and last names (context) and ask the LLM to retrieve $n$ names that start with some letter (query). We can test if the LLM hallucinates by checking whether the each name it generates is in the prompt, and optimize to reduce the corresponding hallucination rate directly.[2]

We seek synthetic tasks $\tau_{\text{syn}}$ that satisfy two desiderata: (i) the LLM hallucinates frequently on the task, and (ii) we can test for hallucination on the task efficiently. The former means optimizing on the synthetic task teaches the model to hallucinate less, while the latter makes optimization tractable.

## 2.3 REDUCING HALLUCINATION WITH SYNTRA

We now present the entire SYNTRA pipeline. SYNTRA defines a synthetic task (Section 2.2), optimizes either the system message or model weights on the synthetic task (with regularization), then transfers the learned system message or model weights to realistic tasks. We detail each step below.

**Optimization.** To reduce the hallucination rate on the synthetic task, SYNTRA either optimizes system message $\phi$ or model weights $\theta$.

*Optimizing the system message $\phi$* is challenging since the set of possible system messages is discrete, precluding continuous optimization with gradient-descent. To circumvent the discreteness of the system messages, we adapt the prefix-tuning method from (Li & Liang, 2021) to learn a continuous postfix to the system message. Specifically, we exploit the fact that while prompts are discrete, the LLM maps them to sequences of continuous embeddings during inference. To optimize the system message, we append a continuous postfix to the embedded discrete system message, then optimize this postfix with gradient descent. Intuitively, the optimized system message provides "high-level instructions" that we expect to extrapolate better. See Appendix A for details.

---

[2]In this case, we add constraints until there is a unique non-hallucinated output (e.g., names are outputted in order as a comma-separated list) to avoid averaging over the combinatorially many non-hallucinated outputs. We could efficiently optimize other definitions (e.g., is any name not on the list) with zeroth order methods.

*Optimizing the model weights $\theta$* is easy since they are continuous; we use standard fine-tuning.

**Regularizing with reference data.** Reducing the hallucination rate on the synthetic task may transfer well out-of-the-box, but could potentially pick up on task-specific spurious attributes—attributes that only appear in hallucinated outputs in the synthetic task, but can appear in correct outputs in general. For example, models may only output newlines when they hallucinate on the synthetic task, but frequently output newlines in general on realistic tasks. Optimizing on the synthetic task may lead the model to never output a newline, which can compromise performance.

To mitigate the effect of these spurious attributes, we optimize the model to preserve its outputs over a reference distribution. Specifically, for reference distribution $\mathcal{D}_{\text{ref}}$ of prompts, original system message $\phi_{\text{og}}$, model weights $\theta_{\text{og}}$, and KL-divergence $\ell$, we define the reference loss $\ell_{\text{ref}}$ as

$$\ell_{\text{ref}}(\phi, \theta; \mathcal{D}_{\text{ref}}, \phi_{\text{og}}, \theta_{\text{og}}) \coloneqq \mathbb{E}_{p \sim \mathcal{D}_{\text{ref}}} \ell(LLM_{\phi,\theta}(p), LLM_{\phi_{\text{og}},\theta_{\text{og}}}(p)). \tag{2}$$

Training on the reference data helps combat spurious attributes that can take new values on that data. While LLMs may hallucinate on the reference data (and potentially learn that the true hallucination features are spurious), we find empirically that this is not an issue (Section 3.2). Optimizing with reference data resembles the PPO-ptx method from Bai et al. (2022), which mixes in pretraining gradients when optimizing a reward model.

**Full loss function.** We optimize convex combinations of the two losses, i.e.,

$$\ell_{\text{opt}}(\phi, \theta; \tau_{\text{syn}}, \theta_{\text{og}}, \mathcal{D}_{\text{ref}}, \alpha) \coloneqq \alpha \ell_{\text{hal}}(\phi, \theta; \tau_{\text{syn}}) + (1 - \alpha)\ell_{\text{ref}}(\phi, \theta, \mathcal{D}_{\text{ref}}; \phi_{\text{og}}, \theta_{\text{og}}), \tag{3}$$

where $\alpha \in [0, 1]$ is a hyperparameter; setting $\alpha = 1$ optimizes on the synthetic task, while $\alpha = 0$ optimizes to preserve the original model. We then use the learned parameters on realistic tasks.

## 3 EVALUATING SYNTRA

We next present an empirical validation of SYNTRA. We describe the setup (Section 3.1), show how SYNTRA reduces the hallucination rate on realistic tasks (Section 3.2), test output quality and other hallucination metrics (Section 3.3), and isolate the impact of the reference data (Section 3.4).

### 3.1 SETUP

**LLMs.** We primarily evaluate SYNTRA on two LLMs: 13B-parameter Vicuna 1.1 (Chiang et al., 2023), and 13B Orca (Mukherjee et al., 2023). Both are fine-tuned from Llama 13B (Touvron et al., 2023a). We also run a subset of our experiments on Llama-2 (Touvron et al., 2023b) (Appendix B.8).

**Optimization details.** We optimize the system message using prefix tuning (Li & Liang, 2021), and the entire LLM with standard fine-tuning. When optimizing with reference data (denoted Synth. + Ref), we set the factor in Equation (3) to $\alpha = 0.5$, and set $\alpha = 0$ for just the synthetic data. We only hyperparameter tune Orca on separate MS MARCO validation data, and reuse the hyperparameters all other Orca tasks and all Vicuna tasks; see Appendix B.1 for compute and hyperparameter details.

#### 3.1.1 REALISTIC TASKS

We study how well SYNTRA reduces hallucination on three realistic LLM use-cases: search-and-retrieve (MS MARCO), meeting summarization (QMSum), and automated clinical report generation (ACI-Bench). Further details for each dataset and full prompts are in Appendix B.2.

**Search and retrieve (MS MARCO).** We first study hallucination in search-and-retrieve applications, where the LLM must generate an answer from retrieved documents. We use MS MARCO as a source of examples (Nguyen et al., 2016), where the task is to answer real-user Bing queries given 10 retrieved passages. For computational tractability, we select 1000 random queries from the MS MARCO validation set that require a long-form response (as labeled in the original dataset).

**Meeting summarization (QMSum).** We next study hallucination in meeting summarization applications, where LLMs are given a meeting transcript and asked to summarize aspects of it. We use the QMSum dataset as a source of examples (Zhong et al., 2021). QMSum contains meeting transcripts and questions (e.g., "*What did the group discuss about budget balancing?*"). We filter the QMSum train set for entries that fit in the LLM context window, for a total of 852 examples.

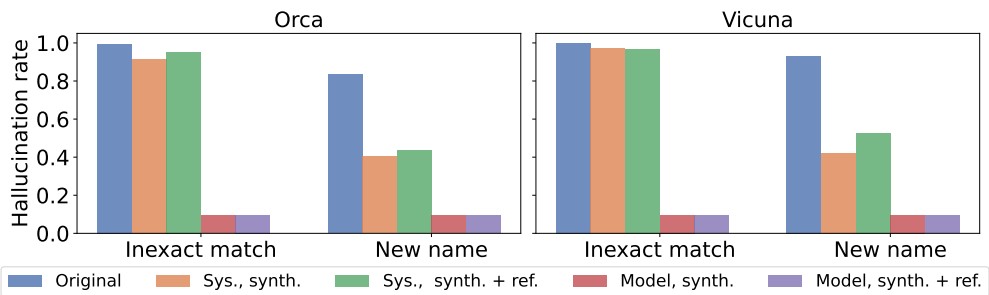

Figure 2: Hallucination rate on the names retrieval task on the original LLM (Original) when optimizing the system message (Sys.) or full LLM weights (Model) on either just the synthetic data (synth.) or mixture of synthetic and reference data (synth. + ref.). We measure the hallucination rate with respect to the exact correct output (inexact match) and the names on the list (new name).

**Automated clinical report generation (ACI-Bench)**. Finally, we study hallucination in automated clinical report generation. We use ACI-Bench (Yim et al., 2023) as a source of examples, which contains dialogs between doctors and patients. Given a dialog, the task is to generate a clinical report with four specific headings. We use all 207 examples from the published dataset.

### 3.1.2 SYNTHETIC TASKS

**Synthetic task (*names retrieval*)**. We define the names retrieval task, where the LLM needs to retrieve certain names from a given list. For example, we might prompt the model with the following:

> **Prompt:**
>
> The following is a list of names
> [Names]
> List the first 5 names where the first name starts with E in the order that they appear. Include both the first and last name in the response. If there are not 5 names that start with E, return all of the names in the list that start with E in the order that they appear.

We generate a dataset of 100,000 examples and test for hallucination by checking whether any name in the output does not appear in the original list. When optimizing to reduce hallucination, to use first-order methods, we optimize the log-likelihood of the unique output allowable by the prompt; in Section 3.2, we show that doing so also reduces the rate of hallucinated names even among outputs that are not exact matches. See Appendix B.3 for details on the dataset construction.

**Other synthetic tasks.** Before trying the names synthetic task, we tried reversing the words in a sentence, and "splicing" pairs of sentences by alternating words from each. We found that while LLMs performed poorly on these, they produced outputs that did not match our intuitive definition of hallucination (e.g., when splicing sentences, LLMs frequently just concatenate the two sentences). We did not try other synthetic tasks; there are likely others that lead to greater gains.

**Reference data**. For the reference data $\mathcal{D}_{\text{ref}}$, we use SQuAD (Rajpurkar et al., 2016) as a source of 50000 prompts. For each passage, we ask the LLM to respond to the associated query, and for half we ask it to explain its reasoning or think step by step. We include details in Appendix B.4.

### 3.2 REDUCING HALLUCINATION WITH SYNTRA

In this section, we measure whether SYNTRA teaches LLMs to hallucinate less. We first verify that SYNTRA reduces the hallucination rate in-domain on the synthetic task, then measure whether it reduces the hallucination rate out-of-domain on realistic tasks.

**Testing in-domain hallucination reduction.** We first want to verify that our names retrieval task satisfies our desiderata for a synthetic task, and that our optimizer works. To do so, we want to verify

| Model | Parameters | Data | MS MARCO | QMSum | ACI-Bench | Average |
|---|---|---|---|---|---|---|
| | *Original Orca* | | *12.2 ± 1.0* | *18.1 ± 1.3* | *47.0 ± 3.5* | *25.8* |
| Orca | Full model | Synthetic | 23.7 ± 1.4 | 24.8 ± 1.5 | 45.4 ± 3.6 | 31.3 |
| | | Synth. + Ref. | 19.6 ± 1.3 | 28.0 ± 1.5 | 47.0 ± 3.5 | 31.5 |
| | Sys. message | Synthetic | **9.9 ± 0.9** | 16.2 ± 1.3 | 44.8 ± 3.5 | 23.6 |
| | | Synth. + Ref. | 10.5 ± 1.0 | **15.8 ± 1.3** | **28.6 ± 3.3** | **18.3** |
| Vicuna | *Original Vicuna* | | *26.2 ± 1.4* | *31.4 ± 1.6* | *50.2 ± 3.5* | *35.9* |
| | Full model | Synthetic | 51.3 ± 1.6 | 56.9 ± 1.7 | 66.3 ± 3.5 | 58.2 |
| | | Synth. + Ref. | **21.1 ± 1.3** | **29.4 ± 1.6** | 50.5 ± 3.5 | 33.7 |
| | Sys. message | Synthetic | 26.2 ± 1.4 | 35.7 ± 1.6 | 83.1 ± 2.6 | 48.3 |
| | | Synth. + Ref. | 24.6 ± 1.4 | 33.3 ± 1.6 | **42.3 ± 3.5** | **33.4** |

Table 1: Hallucination rate (%) measured by GPT-4 across all models, optimized parameters, and tasks (lower is better). We compare against the original model, and optimize the full model or system message, using just synthetic data, or both the synthetic and reference data (SYNTRA).

that the unmodified LLM hallucinates frequently on the names retrieval task, and that optimizing on the names retrieval task reduces hallucination on new in-distribution instances.

In order to test for hallucination on the names retrieval task, we measure whether (i) the generated answer matches the unique correct answer, and (ii) the LLM only generates names on the list.

We report full results in Figure 2, and find that the names retrieval task satisfies our desiderata. Both Vicuna and Orca hallucinate frequently on this task; they are exactly correct less than 1% time, and only generate names on the list less than 17% of the time. Moreover, our optimizer works in-domain; optimizing either the system message or the entire LLM reduces hallucination across all measures.

**Testing hallucination reduction on realistic tasks.** We next test whether reducing hallucination on the names retrieval task reduces hallucination on the realistic tasks from Section 3.1.1. We also study how important the reference data and optimization parameters are for transfer performance.

To measure hallucination on the realistic tasks, we use GPT-4 as an evaluator (OpenAI, 2023), following Yue et al. (2023); Gao et al. (2023a).[3] We prompt GPT-4 with the context, query, and LLM-generated output and ask whether the factual information in the output is grounded up to paraphrases. GPT-4 retrieves any spans that are not grounded, then scores the groundedness of the response from 0 - 10. We say the LLM hallucinates if the returned score is not 10, i.e., there exists some ungrounded span and thus some fabricated content. See Appendix B.5 for further details.

We find that for every realistic task and LLM, SYNTRA is able to reduce hallucination by optimizing on a mixture of the names and reference data (Table 1). On average, optimizing the system message decreases the hallucination rate by 7.5 points on Orca (a 29% reduction), and 2.5 points on Vicuna (a 7% reduction). For specific applications, the decrease can be larger; for example, SYNTRA reduces the hallucination rate on ACI-Bench by 16 points on Orca, and 8 points on Vicuna.

**Additional analysis.** We next discuss which components are necessary for SYNTRA to transfer to realistic tasks, including the optimization parameters, reference, and LLM, in further detail.

*System message vs LLM weights.* Our results demonstrate that optimizing the system message instead of the whole LLM is sometimes necessary to reduce hallucination, even though fine-tuning is strictly better in-domain. On Orca, fine-tuning the full LLM actually *increases* the hallucination rate across most tasks, while optimizing the system message produces consistent reductions. On Vicuna, fine-tuning reduces hallucination on two out of the three tasks, but is slightly worse than optimizing the system message on average.[4] We hypothesize that this gap exists in part because fine-tuning latches onto more spurious attributes of the synthetic task, and provide evidence in Section 3.4.

---

[3] GPT-4 is tractable to use for our evaluation, but would be expensive to use in optimization directly.

[4] Fine-tuned Vicuna reduces hallucination by removing many non-hallucinated entities in addition to hallucinated ones, which reduces the fidelity of the outputs; see Table 3 in Section 3.3 for details.

| Model | Parameters | Data | MS MARCO | | | | QMSum | | | |
|-------|-----------|------|------|-----|-----|-----|------|-----|-----|-----|
| | | | BLEU | R-1 | R-2 | R-L | BLEU | R-1 | R-2 | R-L |
| Orca | *Original Orca* | | *10.5* | *29.4* | *18.6* | *25.3* | *6.2* | *34.3* | *11.1* | *22.6* |
| | Sys. message | Synthetic | 14.4 | 35.0 | 23.2 | 30.9 | 6.1 | 34.0 | 11.2 | 22.9 |
| | | Synth. + Ref. | 13.8 | 33.9 | 22.6 | 29.7 | 6.3 | 34.4 | 11.5 | 22.9 |
| | Full model | Synthetic | 13.9 | 31.6 | 20.4 | 28.8 | 4.0 | 24.0 | 7.8 | 17.1 |
| | | Synth. + Ref. | 11.4 | 30.1 | 19.0 | 26.3 | 5.9 | 33.4 | 11.1 | 22.7 |
| Vicuna | *Original Vicuna* | | *8.7* | *26.0* | *15.5* | *22.1* | *5.9* | *33.8* | *11.0* | *22.6* |
| | Sys. message | Synthetic | 8.5 | 26.6 | 15.4 | 22.8 | 5.5 | 33.1 | 10.8 | 22.4 |
| | | Synth. + Ref. | 10.2 | 27.8 | 17.0 | 23.8 | 5.7 | 33.4 | 11.1 | 22.6 |
| | Full model | Synthetic | 4.9 | 17.7 | 7.9 | 16.3 | 1.8 | 14.8 | 5.0 | 11.4 |
| | | Synth. + Ref. | 11.2 | 29.6 | 18.7 | 25.7 | 5.8 | 33.4 | 10.8 | 22.3 |

Table 2: Comparison between outputs generated using SYNTRA and human-written reference outputs. We abbreviate ROUGE-1, ROUGE-2, and ROUGE-L with R-1, R-2, and R-L. We find that SYNTRA is consistently closer to the reference summaries on MS MARCO than the original model, and is comparable on QMSum. For all metrics, higher is better.

*Regularizing with reference data.* Regularizing with reference data is also critical to reduce the hallucination rate; when optimizing the system message, for 5 out of the 6 task-LLM pairs, adding the reference data reduces the hallucination rate. While the reference data does not significantly impact the hallucination rate when fine-tuning Orca, it reduces the hallucination rate by over 24 points on average when fine-tuning Vicuna. The reference data helps helps break task-specific spurious attributes, which we conjecture is responsible for the improvement (see Section 3.4).

SYNTRA *reduces the hallucination rate more on Orca than Vicuna.* Finally, SYNTRA consistently reduces the hallucination rate more on Orca than Vicuna, and the gains of optimizing the system message instead of the full LLM weights are larger for Orca. This is partly because we only hyperparameter tune using Orca; hyperparameter tuning separately on Vicuna was intractable due to computational constraints, but would likely lead to further gains. This could also be due to a difference in the quality of the underlying LLMs; the original Orca hallucinates much less than Vicuna, and thus might better leverage optimized instructions, without requiring changes to model weights.

## 3.3 ASSESSING HOW SYNTRA REDUCES HALLUCINATION

We next aim to test *how* SYNTRA reduces hallucination, and in particular verify that SYNTRA does not simply exploit shortcomings of the GPT-4 evaluation. To do so, we test whether the SYNTRA-generated outputs are (i) lower-quality and (ii) contain fewer specific details, relative to original outputs. These tests also provide further evidence that SYNTRA reduces the true hallucination rate.

**Testing output quality.** We first aim to test whether SYNTRA reduces output quality by measuring whether its outputs drift away from reference outputs. To do so, we compute the BLEU score (Papineni et al., 2002), and ROUGE-1, -2, and -L scores (Lin & Rey, 2004), which compare the $n$-gram overlap between the LLM generated output and the reference output. We compute these metrics for MS MARCO and QMSum, as they provide high-quality reference outputs.

We include full results in Table 2, and find that SYNTRA's outputs do not drift away from reference outputs. In contrast, when optimizing the system message with reference data, SYNTRA has comparable scores across all metrics and LLMs on QMSum, and actually *increases* all metrics on MS MARCO. These metrics also reveal that outputs GPT-4 labels as hallucinated are indeed lower quality; Vicuna fine-tuned on only synthetic data, which hallucinates the most, has the lowest scores.

*What does this say about hallucination?* These metrics provide an indirect signal on whether LLMs hallucinate less; outputs with hallucinated content should be less similar to the fully-grounded reference outputs.[5] Thus, SYNTRA's consistent improvement on MS MARCO across all automated metrics for both LLMs provides further evidence that it reduces the true hallucination rate.

---

[5]Since hallucinated content does not appear in reference outputs, hallucination tends to decrease similarity.

| Model | Parameters | Data | Ungrounded (⇓) | Grounded (⇑) | %↓ Ungrounded (⇑) | %↓ Grounded (⇓) |
|---|---|---|---|---|---|---|
| Orca | | *Original Orca* | *9.9* | *17.2* | - | - |
| | Sys. message | Synthetic | 7.8 | 16.0 | 21.4% | 6.8% |
| | | Synth. + Ref. | 8.0 | 16.9 | 19.2% | 1.4% |
| | Full model | Synthetic | 2.0 | 5.8 | 79.4% | 66.2% |
| | | Synth. + Ref. | 6.3 | 15.1 | 36.5% | 12.2% |
| Vicuna | | *Original Vicuna* | *9.4* | *17.6* | - | - |
| | Sys. message | Synthetic | 4.7 | 11.2 | 49.3% | 36.5% |
| | | Synth. + Ref. | 6.3 | 15.0 | 32.3% | 14.8% |
| | Full model | Synthetic | 1.4 | 0.2 | 84.9% | 99.0% |
| | | Synth. + Ref. | 5.3 | 14.5 | 43.2% | 17.6% |

Table 3: Entity evaluation between the prompts and outputs in ACI-Bench. Using a NER model, we measure the number of entities that are in the output that do not appear in the prompt (Ungrounded, lower is better), and that do appear in the prompt (Grounded, higher is better), along with the percentage decrease in ungrounded and grounded entities relative to the original output.

**Testing for detail removal.** We next aim to test whether the LLM avoids hallucinating by generating fewer details. To do so, we use a commercial-grade named entity model that is optimized for healthcare (Appendix B.5) to compute all entities in the output and context on ACI-Bench. We then test for details by measuring the number of *grounded entities*, i.e., entities in the output that are also in the context. To test for hallucination directly, we also measure the number of *ungrounded entities*, i.e., entities in the output that do not appear in the context (and thus could be hallucinated). [6]

We include full results in Table 3, and find that SYNTRA does not significantly reduce the number of grounded entities; it decreases the number of grounded entities that Orca generates by 1.4%, and by 14.8% for Vicuna. However, SYNTRA decreases the number of ungrounded entities by much more: by 19.2% for Orca and 36.5% for Vicuna. Fine-tuning and training without the reference data eliminate more grounded and ungrounded entities. See Appendix B.7 for further analysis.

*What does this say about hallucination?* These results provide a direct signal that SYNTRA reduces the hallucination rate by decreasing the number of ungrounded entities across all tested methods.

### 3.4 REFERENCE DATA COMBATS SPURIOUS ATTRIBUTES

We next aim to identify whether optimizing on the reference data combats task-specific spurious attributes, which can drive up the hallucination rate. To do so, we identify newlines as an easy-to-evaluate spurious attribute. In the names retrieval task, all correct answers do not have newlines, so LLMs may learn that newlines are associated with hallucinations. However, never outputting a newline could lead to errors when transferring; for example, LLMs may abruptly end generation before answering the query, rather than outputting a newline (e.g., "*The items are:*" [ends]).

We report the newline rate across all methods and tasks in Table 4 of Appendix B.6, and find that training on reference data helps mitigate the effect of the spurious attributes. Training on the names retrieval task routinely reduces the newline rate, but adding in the reference data recovers much of the drop. We measure the presence of newlines as we identified it as a potential spurious attribute and could measure it easily, but reference data likely similarly combats unknown spurious attributes.

## 4 RELATED WORK

**Hallucination.** We aim to reduce hallucination in text generation systems when all salient information is included in context. Text generation systems frequently hallucinate; see a general survey (Ji et al., 2023a), and surveys restricted to abstractive summarization (Maynez et al., 2020; Huang et al., 2021) for examples. Hallucination is one of many documented potential risks of deploying LLMs (Bender et al., 2021; Bommasani et al., 2021; Weidinger et al., 2021).

---

[6] An alternative metric would be to test what fraction of outputs have *no* ungrounded entities, but this produces false-positives due noise in the entity model. For example, if "prescribe" appears in an output, the entity model extracts "rib", in which case the output is considered hallucinated whenever "rib" is not in the input.

Hallucination is hard to detect automatically. Some work measures hallucination by comparing outputs to reference summaries using BLEU score (Papineni et al., 2002), ROUGE score (Lin & Rey, 2004), or entity overlap (Nan et al., 2021). More recent work measures hallucination by decomposing outputs into atomic facts and evaluating them (Min et al., 2023), or uses LLMs (Yue et al., 2023; Gao et al., 2023a). Another line of work suggests that LLMs may encode whether they are hallucinating within their activations (Kadavath et al., 2022; Burns et al., 2023; Azaria & Mitchell, 2023), but this has not been scaled to abstractive summarization settings.

There are a few classes of methods to reduce hallucination. Some methods adjust the generation process, either by changing the decoding strategy (Tian et al., 2019; Shi et al., 2023), teaching the LLM to cite (Gao et al., 2023b), edit (Gao et al., 2023a), or abstain (Cao et al., 2023) as it generates, or incorporating knowledge graphs (Ji et al., 2023b) and external documents (Mallen et al., 2023). One line of work aims to reduce hallucination with prompting strategies (Jung et al., 2022; Zhou et al., 2023), while another edits model internals to make LLMs more honest (Li et al., 2023), or faithful to the context (Hernandez et al., 2023). The closest line of work to ours trains models to hallucinate less with contrastive learning (Cao & Wang, 2021; Tang et al., 2022) or reinforcement learning (Roit et al., 2023). Our work optimizes to reduce hallucination directly on synthetic data.

Other work optimizes for other LLM behaviors such as helpfulness or harmlessness. They do so by learning a reward function capturing the behavior with human feedback (Sadigh et al., 2017; Christiano et al., 2017), then optimizing LLMs using this reward function (Stiennon et al., 2020; Bai et al., 2022; Ouyang et al., 2022). Such work aims to learn an approximate a general-purpose objective to optimize, while SYNTRA optimizes an exact objective on a narrower domain.

**Synthetic data.** SYNTRA leverages synthetic data to reduce hallucination. Synthetic data generated by LLMs has been used to train higher-quality small models (Eldan & Li, 2023; Gunasekar et al., 2023), train models to follow instructions (Dubois et al., 2023; Chiang et al., 2023; Mukherjee et al., 2023), and make models better at clinical text mining (Tang et al., 2023). A closer line of work to ours trains LLMs with synthetic data that comes from non-LLM sources; Sanh et al. (2021) convert existing NLP benchmarks to tasks for instruction tuning, Wei et al. (2023a) add random synthetic labels, and Wei et al. (2023b) adapt benchmarks to reduce sycophancy. The closest work to ours is Zhang et al. (2023), which aims to characterize hallucination by evaluating it on synthetic tasks.

**Prompt optimization.** SYNTRA reduces hallucination by optimizing the LLM system message; to do so we append a continuous postfix, then optimize using prefix-tuning (Li & Liang, 2021). Both Li & Liang (2021) and Su et al. (2022) demonstrate that prefixes can transfer well between classification tasks; our work provides further evidence of this for generative tasks. Some work aims to optimize a discrete prompt directly to improve performance on classification tasks (Shin et al., 2020; Wen et al., 2023); such methods could in principle be plugged into SYNTRA directly.

## 5 DISCUSSION

We introduce SYNTRA a method to reduce hallucination by defining and exploiting synthetic tasks. SYNTRA reduces the hallucination rate across a suite of realistic evaluation tasks.

There are many natural ways to improve SYNTRA. We could improve the optimization method by searching for better hyperparameters for fine-tuning and prefix-tuning, testing other fine-tuning methods like LoRA (Hu et al., 2022), or jointly optimizing the system prompt and model weights. We could optimize over discrete system prompts, which are harder to optimize over but might generalize better. We could also choose better synthetic tasks; this could include searching for a synthetic task that reduces hallucination more over all tasks, separate synthetic tasks for each general task, or mixtures of synthetic tasks. We expect many of these would further reduce the hallucination rate.

Finally, we discuss the tradeoffs between synthetic tasks and demonstrations. Synthetic tasks allow us to scalably generate lots of data and facilitates optimization, but SYNTRA induces a lossy transfer step. On the other hand, optimizing on demonstrations or preferences eliminates the transfer step and may generalize more broadly, but demonstrations are expensive to obtain, and the optimization objective can be misspecified (Casper et al., 2023). These methods may be complementary — SYNTRA already complements existing methods to reduce hallucination, as Orca and Vicuna are both instruction tuned already (likely reducing hallucination). Understanding the comparative strengths of synthetic tasks and real demonstrations could help us train more reliable and safe LLMs.

ACKNOWLEDGMENTS

We thank Besmira Nushi, Alex Pan, Mert Yuksekgonul, Olivia Watkins, Jacob Steinhardt, and Ruiqi Zhong for feedback on this work.

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

## A    ADDITIONAL DETAILS FOR THE SYNTRA PIPELINE

In this section, we provide additional details on our adaption of Li & Liang (2021)'s prefix-tuning method.

**LLM preprocessing.** We first define some prelimaries for LLM preprocessing. Given a prompt $p$, the language model splits the prompt into tokens with the tokenizer function $t$, i.e., $t(p) = p_1, \ldots, p_m$, where $p_i$ is the $i^{th}$ token of the tokenized prompt. Language models then map each token to a continuous embedding using a simple lookup table, and concatenate them. This produces an embedded prompt $e(p)$ that is a $d \times m$ matrix where $d$ is the dimension of the language model embeddings.

**Appending the postfix**. To add the postfix to the system message, we simply initialize a random $d \times n$ matrix, then append it to the embedded system message to form a $d \times n + m$ matrix.

In practice, the implementations of Orca and Vicuna embed the system message and prompt jointly. To identify where the system prompt ends, we exploit the formatting they use and insert the system prompt right before the first "newline" embedding, ensuring it is placed correctly.

**Optimizing the postfix.** To optimize the postfix, Li & Liang (2021) introduce a lower-dimensional vector that they optimize, then transform back to embedding space with a matrix. Instead, we optimize the postfix in the full embedding dimension, which is 5120 for both variants of Llama that we consider. Li & Liang (2021) do this to stabilize optimization for smaller models; our work suggests that this might not be necessary for larger models.

## B    ADDITIONAL DETAILS FOR EVALUATING SYNTRA

### B.1    OPTIMIZATION, INFERENCE, AND HYPERPARAMETER DETAILS.

In this section, we describe the specific experimental details used.

**Compute details.** We perform all of our experiments on a single NVIDIA A100-PCIE-80GB GPU, except for fine-tuning, for which we use four A100s. We run inference and optimization for all models in bfloat16 precision. We use 13B parameter models, as these were the largest that could fit in memory when optimizing the postfix.

**Inference details.** For all experiments (including the original LLMs with no optimization), we use the system message "*You are a helpful, honest, and conservative AI system designed to answer queries using only the provided context.*". Intuitively, we aim to make the baseline LLMs hallucinate less via prompting, so that the gains made by optimizing do not rediscover easy prompting gains. We sample with temperature 0.7 when generating, and have a max sequence length of 1024 tokens. Since the context length of the models we study is 2048 tokens, prompts that have more than 1024 tokens can only have up to 2048 combined tokens in the prompt and output.

**Prefix tuning details.** We append a $n = 10$ token postfix to the embedded system message, which has 51200 parameters (10 tokens times model embedding dimension 5120). We initialize the postfix to be 10 copies of Llama's "space" token embedding. We optimize the postfix with Adam (Kingma & Ba, 2015), using learning rate 1e-4, no weight decay, epsilon 1e-7 and otherwise the default HuggingFace parameters (Wolf et al., 2019). We chose the learning rate out of two options (1e-3 and 1e-4), and the number of tokens out of three options (1, 10, and 20) based on the hallucination rate using Orca on a separate MS-Marco validation set; we then recycle the same hyperparameters for all other realistic tasks and models. This means that SYNTRA is not hyperparameter tuned on Vicuna at all. We use episilon 1e-7 to avoid numerical stability issues that arise when converting models to bfloat16.

**Fine-tuning details** In order to fine-tune the baseline models Orca and Vicuna 1.1, both of which have 13 billion parameters, we use copy the hyperparameters from (Chiang et al., 2023). Specifically, we use a learning rate of 5e-5, warm up ratio of 0.03, weight decay is 0, and we run fine-tuning for one epoch with batch size of 12 per device on four NVIDIA A100-PCIE-80GB GPUs in bfloat16 precision.

For every method, we optimize for one epoch which is exactly 100000 examples. When optimizing on synthetic data mixed with the reference data, we use 50000 examples on the names task and 50000 examples on the reference task.

## B.2 DATASET DETAILS

We next describe the dataset details for each of the realistic tasks we study.

**MS MARCO.** We first describe how we get our MS MARCO subset, using the data from (Nguyen et al., 2016). We start with the validation set, then filter for examples where response type is a "description" (i.e., requires a long-form response), and where at least one document is useful for producing the answer (to enforce that the model only needs to use the context to answer). These labels are included in the data. Of the remaining examples, we randomly choose 1000.

For MS Marco, we use the following prompt

> **Prompt:**
>
> ### System: You are a helpful, honest, and conservative AI system designed to answer queries using only the provided context.
> ### Human: The following is a list of passages:
> -[Passage 1]
> ...
> -[Passage 10]
> Using the passages, respond to the following query:
> [query]
> ### Assistant:

Where here the passages and the query are from the dataset.

**QMSum.** We next curate a subset of QMSum (Zhong et al., 2021). To do so, we start with the training set, then take all queries from the "specific query list" associated with each meeting. For each query in the list, we filter the meeting transcript based on the "relevant text spans" as our transcript, then only include those that have under 1800 tokens according to the Llama tokenizer (so models can generate at least 248 tokens in response). This produces 852 examples. We use the training set rather than the validation because the validation set has far fewer examples, and we don't optimize against this dataset directly.

For QMSum we use the following prompt

> **Prompt:**
>
> ### System: You are a helpful, honest, and conservative AI system designed to answer queries using only the provided context.
> ### Human: The following is a meeting transcript:
> [relevant lines of meeting transcript]
> Using the transcript, respond to the following query:
> Query: [query]
> ### Assistant:

Where the relevant lines of the meeting transcript and the query are taken directly from the dataset.

**ACI-Bench.** Finally, we describe our adaptation of the ACI-Bench dataset (Yim et al., 2023). We combine the train, validation, and three test splits for a total of 207 examples; this is all the possible examples in the dataset. Each examples contains a dialog between a patient and doctor.

We use the prompt that Yim et al. (2023) use with our system prompt directly. This gives us the following prompt:

> **Prompt:**
>
> ### System: You are a helpful, honest, and conservative AI system designed to answer queries using only the provided context.
> ### Human: Summarize the conversation to generate a clinical note with four sections: HISTORY OF PRESENT ILLNESS, PHYSICAL EXAM, RESULTS, ASSESSMENT AND PLAN. The conversation is: [dialog].
> ### Assistant:

Here, the dialogues are taken directly from the dataset.

### B.3 NAMES DATA

We next describe our names synthetic data. We generate names by taking the top 50000 first and last names in the U.S. from Remy (2021), then from these select 100 random first and last names, then combine them.

> **Prompt:**
>
> ### System: You are a helpful, honest, and conservative AI system designed to answer queries using only the provided context.
> ### Human: The following is a list of names
> [Name 1]
> ...
> [Name 100]
> List the first 5 names where the first name starts with [first letter] in the order that they appear. Include both the first and last name in the response. If there are not 5 names that start with [first letter], return all of the names in the list that start with [first letter] in the order that they appear.
> ### Assistant:

Here, the first letter is randomly chosen among the all letters for which there is at least one name, and the names are randomly generated according to the above procedure.

In Figure 2, we evaluate on 1000 randomly generated names prompts that are separate from the 100000 training examples.

### B.4 REFERENCE DATA

As a source of reference data, we use SQuAD (Rajpurkar et al., 2016). Specifically, we take random SQuAD passages and their associated queries to create the following prompts.

> **Prompt:**
>
> ### System: You are a helpful, honest, and conservative AI system designed to answer queries using only the provided context.
> ### Human: The following is a passage:
> [SQuAD passage]
> Using the passage, respond to the following query
> [query]
> [response type]
> ### Assistant:

Here, SQuAD passage and query come directly from the SQuAD dataset. To get a more diverse set of prompts and outputs, we try different options for [response type]. For 25000 examples we leave it empty, and for 25000 examples we use chain-of-thought style prompts (Wei et al., 2022); for 10000 examples set response type to "While performing the task explain your reasoning", and for 15000 examples we set the response type to a chain-of-thought style prompt: "While performing the task think step-by-step and justify your steps".

| Model | Parameters | Data | MS MARCO | QMSum | ACI-bench | Average |
|---|---|---|---|---|---|---|
| Orca | *Original Orca* | | *12.4* | *4.7* | *88.4* | *35.2* |
| | Sys. message | Synthetic | 6.7 | 2.0 | 86.5 | 31.7 |
| | | Synth. + Ref. | 6.9 | 3.8 | 88.4 | 33.0 |
| | Full model | Synthetic | 0.0 | 0.2 | 16.9 | 5.7 |
| | | Synth. + Ref. | 4.7 | 3.2 | 77.8 | 28.5 |
| Vicuna | *Original Vicuna* | | *6.9* | *2.2* | *83.1* | *30.7* |
| | Sys. message | Synthetic | 2.8 | 0.7 | 28.5 | 10.7 |
| | | Synth. + Ref. | 3.8 | 0.7 | 57.0 | 20.5 |
| | Full model | Synthetic | 0.0 | 0.4 | 22.7 | 7.7 |
| | | Synth. + Ref. | 4.3 | 2.7 | 82.6 | 29.9 |

Table 4: Newline rate (%) when optimizing either the system message or full model on just synthetic or synthetic + reference data, for both Orca and Vicuna. Optimizing on just the synthetic data tends to reduce the newline rate (a spurious attribute), adding the reference data restores it.

## B.5 ADDITIONAL EVALUATION DETAILS

**GPT-4 evaluation.** In this section, we describe how we used GPT-4 as a judge, following Yue et al. (2023). The prompt in Yue et al. (2023) asks GPT-4 if the context supports / contradicts the response, or if there's not enough information. However, we found that using this prompt reported hallucination rates of under 3%, which we empirically found was far lower than the true hallucination rate.

Instead, we prompt GPT-4 to score whether each piece of information in the model response is supported by the context, and return ungrounded spans. We encourage it to check each piece of information from the reply, especially focusing on numbers, dates, names, etc., without worrying about paraphrases. We describe the task of GPT-4 as validating the outputs of another language model. The full prompt is proprietary, and mimics prompts used to test for grounding in production.

**Similarity evaluation.** We next provide additional details for computing the BLUE, ROUGE-1, -2, -and -L scores from figure Table 2. When computing BLEU score, we use the method introduced in (Post, 2018). When there are multiple possible reference summaries available, we choose the last reference summary.

**Entity evaluation.** For retrieval of healthcare entities on the ACI-Bench dataset, we used a Named-Entity Recognition service from Microsoft Azure called **Text Analytics for Health**[7]. In order to reduce noise, we select a subset of medical and regular entity types to be extracted from the documents, and a minimum confidence score of 0.75. The healthcare entities that we select include *Allergen, BodyStructure, ConditionQualifier, ConditionScale, Course, Diagnosis, Direction, ExaminationName, Expression, GeneOrProtein, MedicationClass, MedicationForm, MedicationName, MedicationRoute, MutationType, SubstanceUse, SymptomOrSign, TreatmentName* and *Variant*. We include yet one more non-healthcare entity, *Quantity*, as numerical reasoning is known to be a challenging task that can lead to hallucination more easily (Ji et al., 2023a).

## B.6 ADDITIONAL REFERENCE DATA RESULTS

In this section, we present the newline experiment results. We include the results in Table 4, and find that the newline rate tends to decrease on the synthetic data, but increases again when training on the synthetic and reference data jointly. The newline rate is a proxy for other spurious attributes that are harder to anticipate and measure.

---

[7]Azure Text Analytics for health is one of the pre-built features offered by Azure AI Language Service. It is a cloud-based API service that applies machine-learning intelligence to extract and label relevant medical information from a variety of unstructured texts such as doctor's notes, discharge summaries, clinical documents, and electronic health records. More information available on: https://learn.microsoft.com/en-us/azure/ai-services/language-service/text-analytics-for-health/overview?tabs=ner

| Model | Parameters | Data | MS MARCO | | | | QMSum | | | |
|-------|------------|------|------|------|------|------|------|------|------|------|
| | | | BLEU | R-1 | R-2 | R-L | BLEU | R-1 | R-2 | R-L |
| Llama 2 Chat 7B | Original Llama 2 Chat 7B | | 5.6 | 0.182 | 0.114 | 0.157 | 4.7 | 0.300 | 0.095 | 0.195 |
| | Sys. message | Synth. + Ref. | 6.0 | 0.202 | 0.122 | 0.174 | 4.8 | 0.304 | 0.098 | 0.199 |
| Llama 2 Chat 13B | Original Llama 2 Chat 13B | | 3.9 | 0.154 | 0.090 | 0.130 | 4.8 | 0.307 | 0.097 | 0.199 |
| | Sys. message | Synth. + Ref. | 7.9 | 0.236 | 0.144 | 0.205 | 4.2 | .267 | 0.088 | 0.180 |

Table 5: Comparison between SYNTRA and human-generated reference datasets on Llama 2 7B Chat and Llama 2 13B Chat. We abbreviate ROUGE-1, ROUGE-2, and ROUGE-L with R-1, R-2, and R-L. We find that SYNTRA is consistently closer to the reference summaries on MS MARCO than the original model, and is comparable on QMSum. For all metrics, higher is better.

## B.7 ADDITIONAL ENTITY ANALYSIS

In this section, we provide additional analysis of the entity results from Table 3. In particular, we measure the change in grounded and ungrounded entities on two axes: whether the whole model is fine-tuned, or only the system messages is optimized, and whether the parameters are trained on only synthetic data. Overall, we find that fine-tuning tends to eliminate more grounded and ungrounded entities than optimizing the system message (potentially leading to responses with less content), while adding in reference data tends to recover many of the grounded entities, while adding back a few of the ungrounded entities. We provide additional details below.

**Full model versus system message.** We first compare the entity loss between optimizing the full model or only the system message. We find that optimizing the system message consistently preserves more grounded entities: optimizing the system message decreases the average number of grounded entities by 10.6 less (61.6%, which is a substantial loss) than the full model when optimizing just synthetic data, and 1.2 less than the full model when optimizing with synthetic and reference data jointly (7%, which is far less substantial). In contrast, the number of ungrounded entities decreases by an average of 4.5 more (45%) when training on just synthetic data, and 1.4 (14%) more when training on synthetic and reference data. Overall, this analysis demonstrates that training on the full model tends to reduce the number of both grounded and ungrounded entities by more than just the system message (sometimes at unacceptable levels), indicating that training the full model tends to shorten responses more (and potentially leave out important details) than training the system message.

**Synthetic data versus synthetic + reference data.** We next compare training on synthetic data, compared to synthetic and reference data. For the system message, adding in the reference data adds 2.4 (14%) grounded entities compared to just training on the synthetic data, while only adding 0.9 ungrounded entities back (9%). Similarly, fine-tuning with reference entities adds 11.8 grounded entities (68.8%), while adding 4.1 ungrounded entities (41.1%). These numbers have caveats; they measure the gain in entities over just training on synthetic data, which is more degenerate for fine tuning, yet nevertheless shows that reference data recovers any of the lost grounded entities, at the cost of adding (substantially fewer) ungrounded entities. Changing $\alpha$ to mix in less reference data is a way for practitioners to adaptively change this tradeoff; in our work, we only study $\alpha = 1$ (no reference data) and $\alpha = 0.5$ (equally mixing).

**Warnings about applying variants of SYNTRA.** While variants of SYNTRA offer different tradeoffs between the number of ungrounded and grounded entities removed, our entity evaluation reveals that some should be entirely avoided. In particular, fine-tuning the whole model only on synthetic data (without reference data) tends to lead to massive reductions in grounded entities; it reduces the number of grounded entities by 66% for Orca, and removes nearly all of them (99%) for Vicuna. We think this is because Vicuna overfits to the synthetic task without reference data. Overall, these results highlight the importance of fine-grained hallucination for detecting degenerate solutions, and show that SYNTRA be tweaked to adaptively trace a pareto frontier between reducing grounded and ungrounded entities.

| Model | Parameters | Data | Ungrounded ($\Downarrow$) | Grounded ($\Uparrow$) | %↓ Ungrounded ($\Uparrow$) | %↓ Grounded ($\Downarrow$) |
|---|---|---|---|---|---|---|
| Llama 2 7B Chat | Original Llama 2 7B Chat | | 16.2 | 21.8 | - | - |
| | Sys. message | Synth. + Ref | 12.7 | 20.4 | 21.2 | 6.2 |
| Llama 2 13B Chat | Original Llama 2 13B Chat | | 14.1 | 22.6 | - | - |
| | Sys. message | Synth. + Ref | 7.1 | 14.0 | 49.1 | 38.1 |

Table 6: Entity evaluation between the prompts and outputs in ACI-Bench. Using a NER model, we measure the number of entities that are in the output that do not appear in the prompt (Ungrounded, lower is better), and that do appear in the prompt (Grounded, higher is better), along with the percentage decrease in ungrounded and grounded entities relative to the original output.

## B.8 LLAMA 2 EXPERIMENTS

In this section, we present results on Llama 2-chat 7B and Llama 2-chat 13B (Touvron et al., 2023b). We use all of the same hyperparameters that were selected on Orca, without completing a hyperparameter sweep.

In Table 5 and Table 6 we report the overlap with reference summaries and entity analysis described in Section 3.3. We find comparable results to Orca and Vicuna. We do not run the GPT-4 evaluation from Section 3.2 due to resource constraints.

