# OpenReview forum: "Teaching Language Models to Hallucinate Less with Synthetic Tasks"
_ICLR.cc/2024/Conference — ICLR 2024 poster_

### Official Review · Reviewer_tStg · 2023-10-30

**Soundness:** 2 fair
**Presentation:** 2 fair
**Contribution:** 3 good
**Rating:** 6
**Confidence:** 3

**Summary:**

The paper talks about how to optimize the postfix of system message to make LLM hallucinate less. By introducing a synthetic task – the names retrieval task – the authors create an environment where hallucination occurrences are both frequent and easily traceable. This approach allows for a straightforward and clear identification of hallucinations - when the LLM generates a name not present on the provided list.

**Strengths:**

Originality: Though previous works have characterized hallucination using synthetic tasks, this paper goes a step further by utilizing synthetic data to actively reduce hallucination.

Quality and Significance: The paper's relevance is evident, considering the persistent challenge of hallucination in LLMs. By offering a tangible method to evaluate and optimize against such hallucinations, the paper contributes a valuable tool to the domain.

Clarity: The paper is generally well-written.

**Weaknesses:**

The evaluation of SYNTRA is somewhat limited, as it focuses on only 2 models and 3 realistic tasks. This raises concerns about the method's ability to generalize across diverse models and tasks. Also, the effect on Vicuna seems marginal. This weak effect could limit the practical utility of SYNTRA, especially if similar results persist across other models.


I believe the way the results are presented can be improved. There are some key takeaway messages that are hidden in the table/figure. Important insights, such as the significant reduction in grounded entities generated by Orca and Vicuna (by 66% and 99% respectively) when using SYNTRA without reference data, are buried within tables and figures. For example, this observation implies that the LLMs, in their attempt to avoid hallucination, compromise by producing limited details. Such significant findings should be more prominently highlighted, especially to stress that this setup may not be a viable solution for hallucination reduction.

I find the axis label "Hallucination rate" misleading in Figure 2 - I think the number presented therein should instead be 1-hallucination rate? or did I miss anything?

**Questions:**

1. How is the convergence of the optimization? Is there any constraint on the postfix such as any constraint on the $d\times n$ matrix?

2. Can SYNTRA be added on top of other classes of methods mentioned in section 2 to further reduce hallucination?

3. The authors mentioned at the end of the Related Work that the direct optimization of a discrete prompt might be viable in SYNTRA as well, can they give more discussions about the pros and cons compared to SYNTRA? Will it be more difficult in terms of the optimization?

4. From Figure 2 and Table 1, it looks like when LLM weights are fine-tuned, they are overly optimized for the synthetic task and as such it might ends up hallucinate more on realistic tasks. Is there any possibility that an early-stopping of the fine-tuning process might actually help in finding better parameters for both synthetic and realistic tasks?

Minor:
1. Appendix A, Appending the postfix, end of sentence, should it be $d\times (n+m)$?
2. Do authors believe reducing hallucination of LLMs solely by optimizing postfix has a limit and is there a way to find that limit?

---

> ### Author Response · Authors · 2023-11-17
> **Response to Reviewer tStg**
>
> Thanks for your review and comments! We’re glad you found SynTra a valuable tool, and respond to your questions below.
>
> ---
>
> _Important insights, such as the significant reduction in grounded entities generated by Orca and Vicuna (by 66% and 99% respectively) when using SYNTRA without reference data, are buried within tables and figures_
>
> Sorry for the confusion; **the 66% and 99% reduction in entities are for the fine-tuned model, not the system message**: the numbers for the system message are (1.4% and 14.8% respectively). We also find that this reduction is small for the Llama 2 models, (e.g., the reduction for Llama 2 7B is only 6.2%). Moreover, this reduction is consistently less than the reduction in ungrounded entities, which are what we aim to remove. We hope this assuages your concern, and if it changes your impression of our results that you’ll consider increasing your score.
>
> ---
>
> _The evaluation of SYNTRA is somewhat limited, as it focuses on only 2 models and 3 realistic tasks. This raises concerns about the method's ability to generalize across diverse models and tasks._
>
> Thanks for raising this point; based on this feedback, we ran SynTra on Llama 2 7B chat and Llama 2 13B chat, both with RLHF.  We find that SynTra manages to reduce hallucination in both models under these datasets in a comparable way to the existing models. We include full results in Appendix B.8, but find substantial improvements; for example, SynTra doubles the BLEU score for MS MARCO 13B.
>
> ---
>
> _The authors mentioned at the end of the Related Work that the direct optimization of a discrete prompt might be viable in SYNTRA as well, can they give more discussions about the pros and cons compared to SYNTRA? Will it be more difficult in terms of the optimization?_
>
> Yes, doing the discrete optimization is likely harder (e.g., see https://arxiv.org/abs/2306.15447 which shows that even the best discrete optimizers over tokens struggle), but may produce more in-distribution system messages that generalize more. We have added a note about this to the discussion.
>
> ---
>
> _I find the axis label "Hallucination rate" misleading in Figure 2 - I think the number presented therein should instead be 1-hallucination rate? or did I miss anything?_
>
> Yes you are right; thanks for raising this! We’ve updated the figure.
>
> ---
>
> _How is the convergence of the optimization? Is there any constraint on the postfix such as any constraint on the matrix?_
>
>
> The optimization converges; we’ve added plots in Appendix B.8. There is no constraint on the postfix.
>
>
> ---
>
> _Can SYNTRA be added on top of other classes of methods mentioned in section 2 to further reduce hallucination?_
>
> This is possible; SynTra already reduces hallucination on top of instruction tuning (for Vicuna and Orca), which already reduced hallucination. We believe this is an exciting direction for subsequent work.
>
> ---
>
> _From Figure 2 and Table 1, it looks like when LLM weights are fine-tuned, they are overly optimized for the synthetic task and as such it might ends up hallucinate more on realistic tasks. Is there any possibility that an early-stopping of the fine-tuning process might actually help in finding better parameters for both synthetic and realistic tasks?_
>
>
> This is possible; there are ways to both improve fine-tuning and postfix-tuning. Nevertheless, the broader paradigm of supervising on known tasks then transferring is a promising approach, and developing better ways to transfer (e.g., other fine-tuning methods or beyond) are promising future directions.

---

> > ### Comment · Reviewer_tStg · 2023-11-20
> > **Thank the authors for the explanations and additional results**
> >
> > > Sorry for the confusion; the 66% and 99% reduction in entities are for the fine-tuned model, not the system message: the numbers for the system message are (1.4% and 14.8% respectively). We also find that this reduction is small for the Llama 2 models, (e.g., the reduction for Llama 2 7B is only 6.2%). Moreover, this reduction is consistently less than the reduction in ungrounded entities, which are what we aim to remove. We hope this assuages your concern, and if it changes your impression of our results that you’ll consider increasing your score.
> >
> > I understand that the 66\% and 99\% reduction in entities are for the fine-tuned model, not the system message. My point in the original review is exactly regarding the fine-tuned approach. If we see this much of reduction in entities for the fine-tuned model, then it should be prominently highlighted that users should also be cautious of this approach (if not never use this) and again stresses that this **fine tuning setup** may not be a viable solution for hallucination reduction.
> >
> > > Thanks for raising this point; based on this feedback, we ran SynTra on Llama 2 7B chat and Llama 2 13B chat, both with RLHF. We find that SynTra manages to reduce hallucination in both models under these datasets in a comparable way to the existing models. We include full results in Appendix B.8, but find substantial improvements; for example, SynTra doubles the BLEU score for MS MARCO 13B.
> >
> > Thank you for including the Llama-2 models. From Appendix B.8, I saw the authors had kindly pointed us to Table 5 and 6. While I am still parsing the results, I want to make sure we are on the same page - there is no result on the hallucination rate like Table 1 for Llama-2 models, is that correct? I am interested in seeing the error bar on the numbers for an equivalent Table 1. I was most concerned about the marginal improvement on Vicuna from Table 1, and I would like to see whether this holds true for Llama-2 models as well.

---

> > > ### Author Response · Authors · 2023-11-20
> > > **Thanks for getting back to us**
> > >
> > > Thanks for getting back to us so quickly! With respect to your questions:
> > >
> > > > I understand that the 66% and 99% reduction in entities are for the fine-tuned model, not the system message. My point in the original review is exactly regarding the fine-tuned approach. If we see this much of reduction in entities for the fine-tuned model, then it should be prominently highlighted that users should also be cautious of this approach (if not never use this) and again stresses that this fine tuning setup may not be a viable solution for hallucination reduction.
> > >
> > > That makes sense, thanks for clarifying this! The 66% and 99% numbers are a baseline within a baseline: they come from training on just the names data (without mixed in reference) using fine-tuning (instead of tuning the system message). The numbers with just fine-tuning are 12.2% and 17.6% for Orca and Vicuna respectively (Table 3). These are nonzero, but fine-tuning reduces the number of "ungrounded" entities by much more: 36.5% and 43.2% respectively, which might be an ok tradeoff for practitioners. We're space constrained with the edits we can make in the rebuttal, but will add a discussion about how (i) fine-tuning lowers the entity counts by more (i.e., beyond the "Fine-tuning and training without the reference data eliminate more grounded and ungrounded entities" that is already there) and (ii) fine-tuning without reference data can be degenerate to the discussion in 4.3. We can also preview this in the appendix if it's helpful, and overall think this largely enhances the message by adding additional evidence that optimizing the system message transfers better.
> > >
> > > >Thank you for including the Llama-2 models. From Appendix B.8, I saw the authors had kindly pointed us to Table 5 and 6. While I am still parsing the results, I want to make sure we are on the same page - there is no result on the hallucination rate like Table 1 for Llama-2 models, is that correct? I am interested in seeing the error bar on the numbers for an equivalent Table 1. I was most concerned about the marginal improvement on Vicuna from Table 1, and I would like to see whether this holds true for Llama-2 models as well.
> > >
> > > Yes, there is no result analogous to Table 1: we mention this in the response to reviewers (but should've also propagated it here). This is due to quota limits on GPT-4; we can likely get results for the smaller ACI-Bench dataset if it's helpful by tomorrow, but probably cannot get results for the two larger datasets. Both of the Llama-2 and Vicuna experiments use default hyperparameters from Orca without any adjustment, which is part of the reason the Orca gap is larger (see end of Section 4.2).
> > >
> > > Thanks again for all of your feedback on our work, and please let us know if there's anything else we can do to help!

---

> ### Author Response · Authors · 2023-11-20
> **Thoughts on our rebuttal**
>
> Dear Reviewer tStg,
>
> Thanks again for your review! As we near the end of the discussion period, we wanted to ask if we addressed questions you raised, and especially if our clarifications about the entity removal change your opinion of the paper. We hope you find our responses useful and would love to engage with you further if there are any remaining points.
>
> We understand that the discussion period is short, and we sincerely appreciate your time and help!

---

> ### Comment · Reviewer_tStg · 2023-11-21
> **Thank the authors for the clarifications**
>
> >That makes sense, thanks for clarifying this! The 66% and 99% numbers are a baseline within a baseline: they come from training on just the names data (without mixed in reference) using fine-tuning (instead of tuning the system message). The numbers with just fine-tuning are 12.2% and 17.6% for Orca and Vicuna respectively (Table 3). These are nonzero, but fine-tuning reduces the number of "ungrounded" entities by much more: 36.5% and 43.2% respectively, which might be an ok tradeoff for practitioners. We're space constrained with the edits we can make in the rebuttal, but will add a discussion about how (i) fine-tuning lowers the entity counts by more (i.e., beyond the "Fine-tuning and training without the reference data eliminate more grounded and ungrounded entities" that is already there) and (ii) fine-tuning without reference data can be degenerate to the discussion in 4.3. We can also preview this in the appendix if it's helpful, and overall think this largely enhances the message by adding additional evidence that optimizing the system message transfers better.
>
> Thank you for providing additional explanations. They have given me a clearer understanding of the problem. Yes, I agree that this paper would greatly benefit from more in-depth discussions, as pointed out by the authors. It would be beneficial if the authors could include such discussions in the appendix, at least, to make this paper more complete.
>
> > Yes, there is no result analogous to Table 1: we mention this in the response to reviewers (but should've also propagated it here). This is due to quota limits on GPT-4; we can likely get results for the smaller ACI-Bench dataset if it's helpful by tomorrow, but probably cannot get results for the two larger datasets. Both of the Llama-2 and Vicuna experiments use default hyperparameters from Orca without any adjustment, which is part of the reason the Orca gap is larger (see end of Section 4.2).
>
> I understand the computational difficulties and do not wish to add to the burden. However, if feasible, please consider including the experiment again after the rebuttal period for the paper's completeness.
>
> Finally, to acknowledge authors' effort in including the additional experimental evidence, I have raised my score to 6.

---

> > ### Author Response · Authors · 2023-11-22
> > **Thanks for your feedback!**
> >
> > Thanks so much for your response! We appreciate all of your feedback throughout the process; it continues to improve the work. With respect to the points you raised:
> >
> > > Yes, I agree that this paper would greatly benefit from more in-depth discussions, as pointed out by the authors. It would be beneficial if the authors could include such discussions in the appendix, at least, to make this paper more complete.
> >
> > This makes sense; to start to address this, we added appendix B.9 to a new revision, where we discuss the entity results more in depth. We will work to integrate this discussion, along some additional exposition for the other experiments, into the main body of subsequent versions that are less length constrained.
> >
> > > However, if feasible, please consider including the experiment again after the rebuttal period for the paper's completeness.
> >
> > Will do; we agree that it would improve the paper, and we will include the experiment and integrate the results into the main body.
> >
> > Thanks again for all of your feedback and for engaging with the rebuttal; please let us know if any additional questions come up.

---

### Official Review · Reviewer_C5mJ · 2023-10-30

**Soundness:** 3 good
**Presentation:** 3 good
**Contribution:** 2 fair
**Rating:** 6
**Confidence:** 4

**Summary:**

The paper proposes a method called SynTra to reduce hallucination in LLMs. SynTra designs a synthetic task where hallucination is easy to evaluate -- retrieving names from a given list. The synthetic data was used for soft-prompt tuning or full fine-tuning from Orca or Vicuna. The evaluation of real-world tasks such as MS MARCO, QMSum, and ACI-Bench shows that soft-prompt tuning with both synthetic data + reference data performs the best in general.

**Strengths:**

- The optimized model or system message can be transferred to real-world downstream tasks. Experiments show this reduces hallucination compared to the 13B parameter models Vicuna and Orca on tasks like search-and-retrieve, meeting summarization, and clinical report generation.
- Novel idea of using a synthetic task to easily isolate and optimize against hallucinations.

**Weaknesses:**

- Lack of baselines: The proposed method was applied to existing fine-tuned LLMs like Vicuna and Orca and the experiments only show that they become better than the original LLMs.
However, we do not know how the quality of the SynTra task compared to other datasets.
For example, a more fair comparison would be finetuning two LLMs starting from LLaMA, one on the VIcuna sharegpt data, the other on the SynTra data. I wonder if the effects of the SynTra data are not as much as the Vicuna data, in terms of reducing hallucinations.
- Comparison with the code datasets: [previous work](https://arxiv.org/abs/2210.07128) [1] has found that training the model on code datasets is beneficial for reasoning. I wonder how the quality of the SynTra task compared to directly fine-tuning the model on the code dataset?
- In general, training on the rule-based synthetic data would help is expected. However, the question that matters is how useful this data could be compared to other existing datasets such as instruction tuning datasets or code datasets. I suspect that the effects of synthetic data would be marginal when compared to others.
- The transfer from synthetic to downstream tasks could be brittle: Sometimes full fine-tuning is better than soft-prompt tuning in system messages, but sometimes it's not. Sometimes reference data regularization is helping, sometimes it's not.

[1] "Language Models of Code are Few-Shot Commonsense Learners," Aman Madaan, Shuyan Zhou, Uri Alon, Yiming Yang, Graham Neubig. EMNLP 2022

**Questions:**

- In figure 2, the y-axis says hallucination rate (the lower the better), but according to the passages, the y-axis should be the accuracy (the higher is better). Please clarify this.
- Citation error: (yew Lin & Rey, 2004) -> (Lin & Rey, 2004)

---

> ### Author Response · Authors · 2023-11-17
> **Response to Reviewer C5mJ**
>
> Thanks for your review and comments! We’re glad you appreciated how our method is transferable and novel.
>
> It seems like your main concern is that the reduction in hallucination rate is due to the ShareGPT / original LLM fine-tuning set, rather than SynTra. **We think this is a misunderstanding: we compare SynTra trained on top of the fine-tuned models to the fine-tuned models themselves, not to Llama**. In particular, we show that SynTra reduces hallucination _on top of_ any hallucination reduction obtained by the original fine-tuning. This highlights the complementary nature of SynTra to other approaches; SynTra further reduces hallucination after fine-tuning. We updated all tables to say e.g., “Original Orca”, instead of “Original Model” to emphasize this and hope that if this and our other responses help resolve your concerns, you’ll consider increasing your score.
>
> We address your additional questions below:
>
> ---
>
> _Comparison with the code datasets: previous work [1] has found that training the model on code datasets is beneficial for reasoning. I wonder how the quality of the SynTra task compared to directly fine-tuning the model on the code dataset?_
>
> We haven’t tried fine-tuning on code datasets; it’s possible this would help, although reasoning ability may not directly reduce hallucination. However, we expect that SynTra may be complementary to this approach; SynTra reduces hallucination further after fine-tuning on high-quality GPT-4 generated data for Orca, suggesting it improves performance even on fine-tuned models.
>
> ---
>
> _The transfer from synthetic to downstream tasks could be brittle: Sometimes full fine-tuning is better than soft-prompt tuning in system messages, but sometimes it's not. Sometimes reference data regularization is helping, sometimes it's not._
>
> This is true; for specific datasets some approaches work better than others (e.g., training without the reference data is better on MS MARCO). We think this is a byproduct of transferring the same model weights / system message across multiple diverse tasks, and note that on average, training on the system message + regularization data is best for both models.
>
> ---
>
> _In figure 2, the y-axis says hallucination rate (the lower the better), but according to the passages, the y-axis should be the accuracy (the higher is better). Please clarify this._
>
> Thanks for bringing this up! We updated the figure accordingly.
>
>
> ---
>
> _Citation error: (yew Lin & Rey, 2004) -> (Lin & Rey, 2004)_
>
> Fixed, thank you!

---

> ### Author Response · Authors · 2023-11-20
> **Thoughts on our rebuttal**
>
> Dear Reviewer C5mJ,
>
> Thanks again for your review! As we near the end of the discussion period, we wanted to ask if we addressed questions you raised, and if our clarifications about the experimental setup change your opinion of the paper. We hope you find our responses useful and would love to engage with you further if there are any remaining points.
>
> We understand that the discussion period is short, and we sincerely appreciate your time and help!

---

> > ### Comment · Reviewer_C5mJ · 2023-11-21
> > **Thanks for the response**
> >
> > Thanks for the clarification. I know that SynTra is trained on top of the ShareGPT/Orca fine-tuned models and compared only with the ShareGPT/Orca fine-tuned models. And the improvement exists. I would be willing to raise my score to 6.
> >
> > However, I and still curious about the percentage of improvements achieved by ShareGPT/Orca fine-tuning vs SynTra. For example, ShareGPT/Orca fine-tuning improves to reduce a large amount of hallucinations, while SynTra only reduces a marginal amount of hallucinations. I know SynTra should be less effective as the data is easy to generate compared to ShareGPT/Orca data. But it would be worth discussing how are the effects different between real data and synthetic data.

---

> ### Author Response · Authors · 2023-11-21
> **Thanks for your response**
>
> Thanks so much for getting back to us and for your thoughtful comment! It is indeed probably also the case that the ShareGPT / Orca fine-tuning reduces a lot of hallucinations (and instruction tuning in general is probably required for these tasks, which require cohesive long-form outputs). We didn't run the hallucination evaluation on llama 1 because it seems to struggle with our tasks without instruction tuning, but think exploring the comparative benefits of synthetic and real data on top of the instruction tuned models is interesting, and exciting for future work. The last paragraph in our discussion section discusses the tradeoffs between synthetic data and demonstrations, and we've updated it based on your feedback (see new revision).
>
> Thanks again for engaging in the rebuttal, and please reach out if any other questions come up!

---

### Official Review · Reviewer_MYvf · 2023-10-31

**Soundness:** 3 good
**Presentation:** 3 good
**Contribution:** 3 good
**Rating:** 6
**Confidence:** 3

**Summary:**

The paper proposes a simple method for reducing hallucinations in LLMs: finetuning on a synthetic task where hallucinations are easy to elicit and measure (copying certain entries from a provided list of last names). The authors' best method boils down to optimizing system message via prefix tuning and then using this soft system message generalises well to realistic abstractive summarisation tasks, reducing the hallucination rate (measured in terms of BLEU, number of grounded named entities and GPT-4 judge scores).

**Strengths:**

1. The paper addresses a big open problem (hallucinations in LLMs) and stays up-to-date with state-of-the-art advancements in the field (using benchmarks such as ACI-Bench and models such as Vicuna).
2. I like the general “[just ask for generalization](https://www.notion.so/Teaching-language-models-to-hallucinate-less-with-synthetic-tasks-128f91205a1d45208d3add1298c95f18?pvs=21)” approach to reducing hallucinations that does not use human feedback or human demonstrations directly. One could hope it would scale well as LLM capabilities grow (though the authors don’t show this) and won’t limited by errors and bias of human-provided, on-domain gold demonstrations.
3. The paper is written clearly and was easy to follow.
4. The experiments conducted by the authors are well-designed and comprehensive. For instance, I like how they control for limiting hallucinations by exploiting spurious features in their synthetic task and exploiting the errors of GPT-4 judge and report additional metrics.

**Weaknesses:**

1. The authors only experiments with one model size (13B) and it’s hard to say how well their method scales. Will be benefits of SynTra increase or decrease for finetuned LLaMA 30B or 65B? What about LLaMA 6.5B?
2. The absolute improvements in hallucination rate (Table 1) don’t strike me as very high. I’m not sure they justify the software complexity of soft system message optimization.
3. Relatedly, it would be good to compare SynTra to some simpler baselines, e.g. finetuning on gold labels for some other summarization dataset (e.g. OpenAI’s cleaned Reddit data). I’d also be curious to see how RLHF’d models (e.g. LLaMA-2-*-chat series) do on those benchmarks.

**Questions:**

1. It was a mildly surprising finding for me that soft system message optimisation is so much better than supervised finetuning. Have you tried manipulating the $\alpha$ hyperparameter to increase regularisation for supervised finetuning? I could imagine this being helpful.
2. Is the KL in eq. 2 forward (KL(base, new)) or reverse (KL(new, base))?
3. It’s confusing that “hallucination rate” in Fig. 2 is actually non-hallucination rate (higher is better) while for “hallucination rate” in Table 1 lower is better.

---

> ### Author Response · Authors · 2023-11-17
> **Response to Reviewer MYvf**
>
> Thank you for your feedback on our work! We appreciate that you liked our high level approach and our experimental design, and thought that circumventing human preferences / demonstrations was promising.
>
> You highlighted two places to expand the experiments: we only experimented with one model size, and it would be nice to compare to other baselines (e.g., running on Llama 2). Based on this feedback, we **ran SynTra on Llama 2 7B chat and Llama 2 13B chat, both with RLHF**.  We find the following:
>
>
> * Prior to using SynTra, both Llama 2 chat models hallucinate more (under the automated metrics), than the non-RLHF Vicuna and Orca models.
> * SynTra manages to **reduce hallucination** in both models under these datasets in a comparable way to the existing models. We include full results in Appendix B.8, but find substantial improvements; for example, SynTra doubles the BLEU score for MS MARCO 13B, and always reduces the number of ungrounded entities on ACI-Bench by substantially more than the grounded entities.
>
> We respond to your additional questions below.
>
> ---
>
> _The authors only experiments with one model size (13B) and it’s hard to say how well their method scales. Will be benefits of SynTra increase or decrease for finetuned LLaMA 30B or 65B? What about LLaMA 6.5B?_
>
> SynTra performs comparably on Llama 2 7B chat and Llama 2 13B chat. We do think that SynTra provides an interesting alternative approach to human feedback, in the limit where human feedback becomes harder to elicit (e.g., in higher quality outputs, humans may increasingly struggle to identify hallucinations).
>
> ---
>
> _Relatedly, it would be good to compare SynTra to some simpler baselines, e.g. finetuning on gold labels for some other summarization dataset (e.g. OpenAI’s cleaned Reddit data). I’d also be curious to see how RLHF’d models (e.g. LLaMA-2-chat series) do on those benchmarks._
>
> On the automated metrics, Llama-2-chat seems to perform _worse than both Vicuna and Orca_, although SynTra manages to regain some of the lost performance. We hypothesize that this might be a negative side-effect of the RLHF; humans may prefer responses that embellish slightly, or add ungrounded content in order to be helpful. We see evidence of this in the new ACI-bench entity evaluation of Llama 2; Llama 2 chat models tend to generate far more entities than Orca and Vicuna (Appendix B.8). We think exploring this further would be interesting for subsequent work.
>
> In terms of fine-tuning on gold summarization labels, we think this is out of scope; these labels require expensive human annotations that SynTra seeks to avoid. Nevertheless, we hypothesize that these approaches are complementary. SynTra already demonstrates improvements on top of models that have been instruction tuned with high-quality data (e.g., Orca is fine-tuned on process-based feedback generated by GPT-4), and we expect that it could further improve performance even after models have been fine-tuned on other gold data.
>
> ---
>
> _The absolute improvements in hallucination rate (Table 1) don’t strike me as very high. I’m not sure they justify the software complexity of soft system message optimization._
>
> For some of the models and datasets the change in hallucination rate isn’t high, but for others it is: for example, in ACI-Bench, SynTra reduces the hallucination rate from 47 -> 29, and on Vicuna from 50 -> 42. We also think there are other ways of improving SynTra in practice; while we opt for generality by making a single system message that generalizes across different realistic tasks, practitioners could design synthetic tasks that are “closer” to a specific practical task they want to transfer to, or hyperparameter tune further. We believe this is an exciting direction for subsequent work.
>
> ---
>
> _It’s confusing that “hallucination rate” in Fig. 2 is actually non-hallucination rate (higher is better) while for “hallucination rate” in Table 1 lower is better._
>
>
> Updated the Figure: thanks for raising this!
>
> ---
>
> _It was a mildly surprising finding for me that soft system message optimisation is so much better than supervised finetuning. Have you tried manipulating the $\alpha$ hyperparameter to increase regularisation for supervised finetuning? I could imagine this being helpful_
>
> This is a good question; we only test two values for $\alpha$ (either 1 and 0.5), but it is possible that this, or other interventions, might improve fine-tuning (just as similarly, regularization + optimization \alpha could help SynTra). Intuitively, we expect that the system message helps because it optimizes over “high-level instructions’ that are likely to transfer, instead of actually changing the model mechanism. However, we note that better fine-tuning might help (see second paragraph of section 5), and think this is an interesting direction for subsequent work.
>
> ---
>
> _Is the KL in eq. 2 forward (KL(base, new)) or reverse (KL(new, base))?_
> It should be KL(base, new) — thanks for catching this.

---

> ### Author Response · Authors · 2023-11-20
> **Thoughts on our rebuttal**
>
> Dear Reviewer MYvf,
>
> Thanks again for your review! As we near the end of the discussion period, we wanted to ask if we addressed questions you raised, and if our additional experiments help assuage some of your concerns. We hope you find our responses useful and would love to engage with you further if there are any remaining points.
>
> We understand that the discussion period is short, and we sincerely appreciate your time and help!

---

> > ### Comment · Reviewer_MYvf · 2023-11-21
> > **Thanks for the response**
> >
> > Thanks for the response. I appreciate additional experiments, especially running SynTra on Llama 2 7B chat and Llama 2 13B chat.

---

### Author Response · Authors · 2023-11-17
**Response to all reviewers**

We’d like to thank all of the reviewers for their feedback on our work. Reviewers said our work “addresses a big open problem” [tStg, MYvf], is a “novel idea” [C6mj], “contributes a valuable tool” [tStg], “stays up to date with state-of-the art advances in the field” [MYvf], and is well-written [MYvf, tStg]. Reviewers also appreciated how the learned system message is transferable [C6mj], and that we manage to reduce hallucination without “[requiring] human feedback / demonstrations directly” [MYvF].

Multiple reviewers suggested that they would like to see further experimental validation of SynTra.  Based on this feedback, we **ran SynTra on Llama 2 7B chat and Llama 2 13B chat**. Under the entity and comparisons to the reference summaries, we find that SynTra manages to reduce hallucination in both models under these datasets at a comparable level to Orca and Vicuna. We include full results in Appendix B.8, but find that SynTra can substantially reduce hallucination on these models; for example, SynTra doubles the BLEU score for MS MARCO on Llama 2 13B. Running the GPT-4 evaluation is slower, but we can try to run the GPT-4 evaluation on the smaller datasets (e.g., ACI-BENCH) for the rebuttal if it helps, and will include the full GPT-4 evaluation in subsequent versions of this paper.

We address other reviewer questions and feedback in the individual responses.

---

### Meta-Review · Area_Chair_vMv6 · 2023-12-07

**Metareview:**

The paper introduces a new method for reducing hallucinations in Large Language Models (LLMs). It involves training an LLM to minimize hallucinations on a simple synthetic task, where hallucinations can be easily characterized. This training is then transferred to real tasks. The training process includes prefix tuning at the level of a system prompt, i.e., at a general level for task definition, coupled with a regularization criterion. This simple strategy enables the trained system to transfer effectively from artificial to real hallucination reduction tasks.

All the reviewers agree on the originality and simplicity of the approach. They acknowledge its efficiency on various evaluation tasks. They requested additional comparisons, such as on larger LLM models, which were provided by the authors during the rebuttal, along with additional clarifications. This addressed the main questions raised by the reviewers. Therefore, I propose acceptance.

**Justification For Why Not Higher Score:**

The main strength of the paper as highlighted by the reviewers is the originality of the proposed approach. However  there is still place for improvements.

**Justification For Why Not Lower Score:**

All the reviewers agree on acceptance

---

### Decision · Program_Chairs · 2024-01-16

Accept (poster)